# Microbial biofilm formation and degradation of octocrylene, a UV absorber found in sunscreen

Marcel Suleiman[1,4], Carola Schröder[1,4], Michael Kuhn[2], Andrea Simon[3], Alina Stahl[3], Heike Frerichs[3] & Garabed Antranikian[1]*

Octocrylene is a widely used synthetic UV absorber of sunscreens and found in several environments. Ecological consequences of the accumulation of UV filters are widely discussed. This is the first report revealing the microbial potential to transform octocrylene. A microbial community comprising four bacterial species was enriched from a landfill site using octocrylene as carbon source. From these microorganisms *Mycobacterium agri* and *Gordonia cholesterolivorans* were identified as most potent applying a new "reverse discovery" approach. This relies on the possibility that efficient strains that are already isolated and deposited can be identified through enrichment cultures. These strains formed massive biofilms on the octocrylene droplets. GC-MS analysis after cultivation for 10 days with *M. agri* revealed a decrease in octocrylene concentration of 19.1%. LC-MS/MS analysis was utilized in the detection and quantification of transformation products of octocrylene. *M. agri* thus represents an ideal candidate for bioremediation studies with octocrylene and related compounds.

---

[1] Institute of Technical Microbiology, Hamburg University of Technology (TUHH), 21073 Hamburg, Germany. [2] Beiersdorf Aktiengesellschaft, 20245 Hamburg, Germany. [3] Central Laboratory of Analytical Chemistry, Hamburg University of Technology (TUHH), 21073 Hamburg, Germany. [4] These authors contributed equally: Marcel Suleiman, Carola Schröder. *email: antranikian@tuhh.de

Octocrylene (2-ethylhexyl 2-cyano-3,3-diphenylacrylate) is a synthetic organic filter widely used as UV-absorber (in the UVB region) in sunscreens and personal care products[1]. Recently, several studies focused on the extensive use and potential ecological consequences of octocrylene, since the compound was detected in various water and sediment samples in wastewater[2] oceans[3], lakes, and rivers[4,5]. Moreover, octocrylene was found in the liver tissue of Franciscana dolphins and accumulation of octocrylene in zebrafish was reported upon exposure[6,7]. Further, octocrylene was suggested to potentially affect transcription of genes in the zebrafish's brain and liver[8,9]. So far, little is known about potential degradation and biotransformation of octocrylene by microorganisms. Hence, this is the first study, to our knowledge, which reports on microbial growth in the presence of octocrylene and its degradation and transformation to other compounds.

## Results

Sediment and water samples were taken from an on-site sewage plant and soakage at a landfill site in Singhofen (Germany). Samples were subsequently used for enrichment cultures (inoculation 1% (v/v)) with 0.35% (v/v) octocrylene as carbon source. After 14 days, grown cultures were transferred into fresh medium with two subsequent streakings. Repeatedly, biomass formation on the octocrylene droplet surface could be observed and the medium became turbid compared to the controls. In order to investigate the community composition of the culture, DGGE analysis was performed[10]. The microbial community of the OC-colonizing enrichment culture consisted of four genera, and the distinct bands of the DGGE were obtained and assigned to the bacterial genera *Gordonia* (100% identity), *Mycobacterium* (100% identity), and *Hydrogenophaga* (99% identity) after excision of the respective bands and sequencing (Fig. 1).

Interestingly, *Mycobacterium* and *Gordonia* species have been reported to be associated with degradation of polluting polycyclic aromatic hydrocarbons (PAHs), such as pyrene or oil, and were proposed for application in environmental bioremediation[11–14]. Although different techniques were performed, isolation of pure strains from the enrichment culture was not possible and resulted

in mixed cultures. Therefore, the "reverse discovery" approach was integrated. This approach is based on taking use of already isolated and deposited pure strains, showing high identities to the identified 16S rRNA genes amplified from the organisms of the enrichment culture. Highly identical strains can be purchased from culture collections, in order to identify the most efficient strains capable of degrading octocrylene. Likewise, several deposited strains at the German Culture Collection (DSMZ) belonging to the genera *Mycobacterium* and *Gordonia*, whose 16S rRNA gene showed high identities to the DGGE-identified ones, were tested for their ability to grow in modified DSM media 645 and 65 at 37 °C and 30 °C in the presence of 0.35% (v/v) octocrylene. Interestingly, *M. agri* (DSM 44515) and *Gordonia cholesterolivorans* (DSM 45229) were able to colonize the octocrylene droplet rapidly forming massive biofilms on its surface (Fig. 2).

*Mycobacteria* and *Gordonia* species have been already described to attach to different surfaces that are composed of biomaterials and synthetic compounds initiating the synthesis of an extracellular matrix required for biofilm formation[11,12]. Since carbon sources can influence biofilm formation[11], different concentrations of glucose and glycerol in the growth medium were tested. Remarkably, while reduction of carbon sources had no positive influence on biofilm formation of *G. cholesterolivorans*, *M. agri* showed fast and massive biofilm formation when carbon sources were significantly reduced (0.05% glucose, 0.05% glycerol). Interestingly, in absence of external carbon sources, no biofilms were formed at all, indicating the necessity of small amounts of accessible energy sources to colonize the octocrylene droplet.

Hence, in order to study transformation of octocrylene by *M. agri* and *G. cholesterolivorans*, gas chromatography-mass spectrometry (GC-MS) and liquid chromatography-tandem mass spectrometry (LC-MS/MS) analyses were performed with the pure cultures grown with 0.35% octocrylene (v/v). Control experiments were conducted with *M. agri* and *G. cholesterolivorans* grown in the absence and presence of octocrylene. In addition, the medium containing octocrylene was incubated under the same conditions without inoculation. After 10 days of cultivation, triplicates of cultures and controls were used for GC-MS and LC-MS/MS analyses. Samples were prepared by extraction of the culture using hexane/dichloromethane (1 + 2),

**Fig. 1** Biofilm formation on octocrylene droplet in the enrichment culture. **a** Chemical structure of octocrylene. **b** Observation of biofilms on octocrylene droplets in the enrichment culture of landfill site samples after two transfers. The incubation temperature was 30 °C. Red arrow: biofilm on the octocrylene droplets swimming in the medium; blue arrow: biofilm on octocrylene on the medium surface. Scale represents 1 cm. **c** Diversity analysis of the enrichment culture using DGGE. Used primers: 314 F and 907 R. DGGE was performed at 100 V for 17 h.

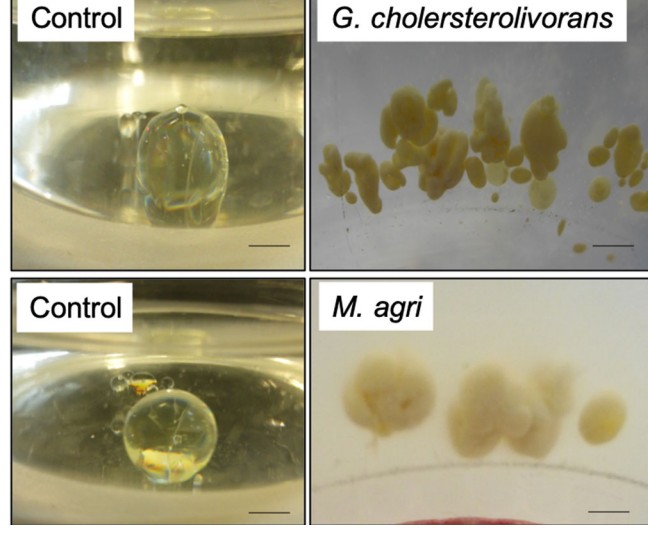

**Fig. 2** Observation of the biofilm formation on the octocrylene droplet after 10 day-incubation with *G. cholesterolivorans* at 30 °C and *M. agri* at 37 °C. Scale represents 0.5 cm.

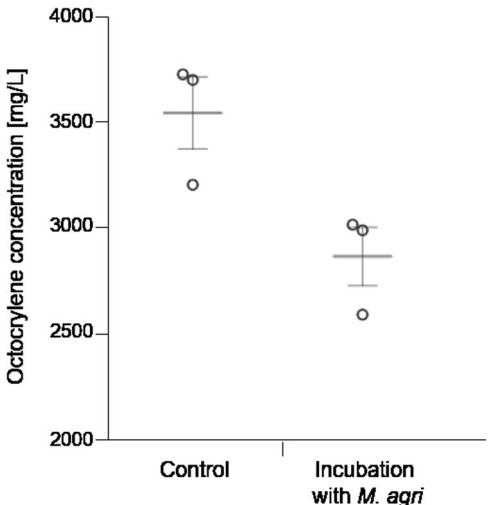

**Fig. 3** Octocrylene degradation assay with *M. agri*. 0.35% octocrylene was incubated in liquid medium without *M. agri* (controls, n = 3) and in presence with *M. agri* (incubation, n = 3) for 10 days. A decrease of 19.1% octocrylene was detected within the incubation time. Mean is shown in gray color, error bars are representing standard error.

for LC-MS/MS measurements the GC extract was evaporated to dryness and refilled with acetonitrile, and the amount of octocrylene was determined using the GC-MS-SIM mode, with $m/z$ 249 as the quantifying ion. The qualitative recording of products (product scan) was done by GC-MS-TIC mode. Separation was performed on an Agilent DB5MS ultra inert capillary column.

In the presence of *M. agri*, the concentration of octocrylene measured by GC-MS-SIM mode decreased from 3536 ± 239 mg/L (controls) to a final concentration of 2861 ± 193 mg/L at day 10, representing a decrease of 19.1% octocrylene (Fig. 3; Supplementary Table 1).

In order to identify possible octocrylene metabolites produced by *M. agri*, analytical standards of 2-(Carboxymethyl)butyl-2-cyano-3,3-diphenylacrylate (DOCCA), 2-Cyano-3,3-diphenyl-lacrylic-acid (CPAA), and 2-Ethyl-5-hydroxyhexyl-2-cyano-3,3-diphenylacrylate (5OH-OC) were synthesized and provided by Bury et al.[9]. Consequently, we were able to compare GC-MS spectra obtained in this work with the metabolite's GC-MS spectra and LC-MS/MS which was subsequently used to quantify the metabolites in our experimental design. While the metabolites DOCCA, CPAA, and 5OH-OC were identified in the MS spectra and quantified via LC-MS/MS, one occurring peak in GC-MS run (Fig. 4, ret. time 15.32 min) could not be identified when comparing to other reported metabolites. The MS spectra of the occurring peak could also not be identified when comparing to MS spectra of published putative metabolites of octocrylene[15].

LC-MS/MS revealed the occurrence of DOCCA with 165 ± 60 µg/L, of CPAA with 11.8 ± 2.6 µg/L and of 5OH-OC with 246.0 ± 34 µg/L. This indicates that *M. agri* has transformed octocrylene to the respective metabolites. The concentration of the detectable metabolites, however, does not correlate with the reduction in octocrylene concentration. It is very likely that the products formed in the medium by *M. agri* (DOCCA, CPAA, and unknown metabolites) were further metabolized by the microorganism and utilized for growth. Non-biotic factors responsible for the reduction of octocrylene can be excluded, since controls without *M. agri* did not show any decrease in octocrylene concentration.

Interestingly, the analysis of growth experiments with *G. cholesterolivorans* revealed no change of octocrylene concentration during incubation, although biofilm was observed. This indicates

that in this case biofilm formation does not go automatically along with transformation or degradation of octocrylene.

## Discussion

Since UV filters such as octocrylene accumulate in nature, the identification of further octocrylene degrading microorganisms is desirable. It has been reported that marine environments are endangered by chemicals and UV-filters of sunscreen[3] and solutions for this pollution have to be investigated. Man-made biofilms were proposed previously for bioremediation of sewage effluent contaminated with hydrocarbons[16]. *M. agri* represents a potential candidate for bioremediation, since it belongs to the rapidly growing *Mycobacteria* and can be found ubiquitously in soil and water systems[13,17,18]. Therefore, a novel eco-friendly application using *M. agri* or specific microbial consortia could be investigated and applied for degradation of chemical UV filter like octocrylene. This work highlights the potential of heterotrophic microorganisms to transform octocrylene in lab experiments. Due to the omnipresence of *M. agri* in the environment, it is conceivable that *M. agri* will also be able to transform the UV filter octocrylene in nature. Hence, further analysis should be conducted regarding biotransformation of octocrylene by microorganisms in natural sediments and waters. Future studies will reveal the metabolic pathway for octocrylene degradation and the key enzymes involved in this process.

## Methods

**Sampling and enrichment cultures**. Sediment and water samples were taken from an on-site sewage plant and soakage at a landfill site in Singhofen (Germany). The samples were subsequently used for enrichment cultures (volume 20 mL) using oxic medium (0.5 g/L NaCl, 1.0 g/L KCl, 0.1 g/L MgSO$_4$, 0.1 g/L MgCl$_2$, 0.5 g/L NH$_4$SO$_4$, 0.5 g NH$_4$NO$_3$, 0.1 g/L CaCl$_2$, 40 mM phosphate buffer, trace element and vitamin solution 141, and 0.005% (w/v) yeast extract). Incubation bottles were inoculated with 1% (w/v) of the soil sample and supplemented with 0.35% (v/v) octocrylene as carbon source. After 14 days of incubation, grown cultures were transferred into fresh medium with two subsequent streakings.

**DGGE analysis**. DGGE analysis was performed as described previously.[19] Amplification of bacterial 16 S rRNA genes was performed using the forward primer 314 F:5′CGCCCGCCGCGCCCCGCGCCCGTCCCGCCGCCCCCGCC CGCCTACGGGAGGCAGCAG-3′ and 907 R: 5′-CCGTCAATTCMTTTGAG TTT-3′).[10] DGGE was performed by employing a DcodeTM System (Bio-Rad Laboratories GmbH, Munich, Germany). A 6% polyacrylamide gel with a denaturing gradient of 35–70% formamide was used and electrophoresis ran for 17 h, at 55 °C and 100 V. The gel was stained in Roti-Gel Stain (Carl Roth GmbH, Karlsruhe, Germany) and afterwards analyzed with a UV transilluminator. In addition, selected bands were cut out and DNA was extracted from the gel by incubation in 10 µL sterile water over night at 4 °C, amplified by PCR and amplificated were sequenced (Eurofins Genomics).

**Cultivation of *Mycobacterium* sp. and *Gordonia* sp**. Several *Mycobacterium* sp. and *Gordonia* sp. showing within their 16S rRNA genes high similarities compared to the sequenced and identified fragments of the DGGE analysis were chosen and ordered from the DSMZ. *Mycobacterium agri* (DSM 44515), *M. aichiense* (DSM 44147) and *M. aromaticivorans* (DSM 45407) were incubated in DSMZ medium 645 (per liter: 5 mL glycerol, 0.5 g (NH$_4$)$_2$SO$_4$, 1.5 g KH$_2$PO$_4$, 1.5 g Na$_2$HPO$_4$, 0.4 g Na-citrate, 0.025 g MgSO$_4$, 0.5 mg CaCl$_2$, 1 mg ZnSO$_4$, 1 mg CuSO$_4$, 0.5 g L-glutamic acid, 0.04 g ferric ammonium citrate, 1 mg pyridoxine, 0.5 mg biotin, 0.05 g oleic acid, 0.85 NaCl, 5 g BSA, 2 g glucose) supplemented with 0.35% octocrylene at a temperature of 37 °C. *Gordonia caeni* (DSM 45852), *G. cholesterolivorans* (DSM 45229), and *G. defluvii* (DSM 44981) were incubated in DSMZ medium 65 (per liter: 4 g glucose, 4 g yeast extract) supplemented with 0.35% octocrylene at a temperature of 30 °C.

**Medium modification of *M. agri* and *G. cholesterolivorans***. The medium of the positive octocrylene-colonizing strains *M. agri* and *G. cholesterolivorans* was modified in order to enhance the biofilm formation. Therefore, the carbon sources of the medium 645 of *M. agri* were highly decreased to 0.05% glucose and 0.05% glycerol and BSA was also reduced to 0.05%. The carbon sources of medium 65 of *G. cholesterolivorans* were decreased to 1 g/L glucose and 1 g/L yeast extract (no addition of malt extract). Incubation was performed in the presence of 0.35% octocrylene.

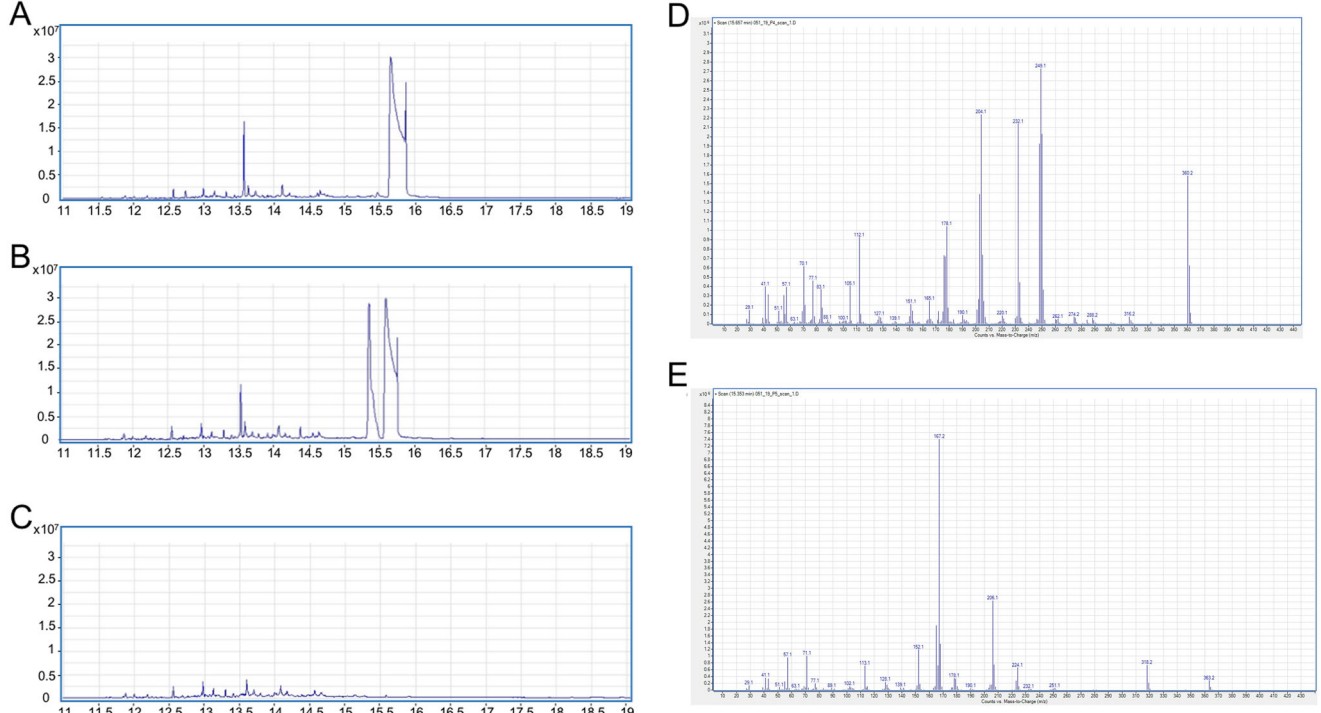

Fig. 4 GC-MS scans (**a–c** TIC mode, cutout, **d–e**: Mass spectra) of the incubation experiment with *M. agri*. **a** Control sample 0.35% octocrylene (rt 15.58 min) incubated for 10 days without *M. agri*. **b** Sample 0.35% octocrylene (rt 15.58 min) incubated for 10 days with *M. agri*. New metabolite was detected (rt 15.32 min). **c** Control sample *M. agri* incubated for 10 days without 0.35% octocrylene. **d** Mass spectrum of the peak identified as octocrylene. **e** Mass spectrum of the unknown metabolite peak (rt 15.32 min).

**Chemicals and materials for analytical approaches**. Reference standard octocrylene was purchased from Sigma Aldrich Ltd, Saint Louis, MO, USA (≥98%). Reference substances 2-(Carboxymethyl)butyl-2-cyano-3,3-diphenylacrylat (DOCCA), 2-Cyano-3,3-diphenylacrylic acid (CPAA) and 2-Ethyl-5-hydroxyhexyl-2-cyano-3,3-diphenylacrylate (5OH-OC) were provided by V.N. Belov, Max Planck Institute for Biophysical Chemistry, Göttingen, Germany. Preparation and purity (>95%) are well described in ref. [9]

Hexane and dichloromethane were obtained from Merck KGaA, Darmstadt, Germany. Acetonitrile (LC-MS grade) was obtained from Merck, Darmstadt, Germany, formic acid (additive for LC-MS) was purchased from Honeywell Fluka, Muskegon, USA.

All reagents were of analytical grade. Deionized water was purified using a Milli-Q-system (Millipore,USA).

**Sample preparation**. The whole incubation culture of *M. agri* and *G. cholesterolivorans* (and controls) (20 mL) was transferred to a centrifuge tube and 10 mL hexane/dichloromethane (1 + 2) was added. After 30 min stirring the sample was centrifuged (6000 rpm). The lower layer was stirred with NaSO₄ and centrifuged (13,000 rpm) again. For GC-MS analysis the liquid was filled in a microbial. For LC-MSMS analysis 1 mL was filled in a volumetric flask, evaporated and refilled with acetonitrile.

**GC-MS analyses**. Quantification of octocrylene and qualitative recording of products: The quantitative determination of octocrylene and qualitative recording of the products were carried out on an Agilent Technologies GCMS system 5977 A MSD with Gerstel KAS4 injection system. The mass selective detector (MSD) was operated in EI mode. All system operations were controlled by MassHunter B.07.001413. Separation and quantification were performed on an Agilent DB5MS ultra inert capillary column, 30 m × 0.25 mm × 0.25 μm film thickness. Sample volumes of 1 μL were injected in split mode (1:5) at 1 mL/min flow. GC system inlet and MS interface temperatures were set at 70 and 280 °C, respectively. The column temperature was programmed as follows: held at 70 °C for 2 min, then ramped up to 140 °C at 10 °C/min, held for 3 min, then to 300 °C at 100 °C/min. The total runtime was 17,6 min. The carrier gas was helium with constant flow of 1 mL/min, the MSD operating parameters were routinely set by tune file. EM potential was set at +1352 V, MS source temperature at 230 °C and the quadrupoles at 150 °C.

Quantitative determination of octocrylene was done by SIM mode, with quantifying ion with $m/z$ 249 and qualifying fragment ions $m/z$ 360 and 204. Recording of products was done by TIC mode (from $m/z$ 4–450, solvent delay 5 min, dwell time 0.2 ms).

**Table 1 Characteristics of metabolites of octocrylene tested in this study.**

| Analyte | Polarity | Q1 | Q3 |
|---|---|---|---|
| DOCCA | Positive | 364 | 250 |
| | | | 176 |
| CPAA | Negative | 248 | 204 |
| | | | 203 |
| 5OH-OC | Positive | 378 | 232 |
| | | | 105 |

**Preparation of analytical standards (GC-MS)**. Stock solution was prepared with a concentration of 5300 mg/L using dichloromethane as solvent. By diluting the stock solution with hexane-dichloromethane (1 + 2) calibration standards were prepared with levels from 5,0 mg/L to 200 mg/L. While the stock solution was stored at −18 °C, the calibration standards had to be newly made for each analytical run.

**LC-MSMS analyses**. Instrumentation and parameters: The quantitative determination of metabolites was carried out on an Agilent 1260 HPLC system (Agilent Technologies Deutschland GmbH and Co. KG, Waldbronn, Germany), equipped with binary pump, degasser, thermostat autosampler, column oven, which is coupled to the Sciex Qtrap 5500 tandem mass spectrometer (AB Sciex Germany GmbH, Darmstadt, Germany).

Separation was achieved by using a Phenomenex Synergi Fusion RP 80 A column (4 μm, 50 × 2 mm) with a Phenomenex standard C18 guard column. The mobile phase consisted of Milli-Q water with 0.1% formic acid (A) and acetonitrile with 0,1% formic acid (B), the flow rate was set to 0.7 mL/min. The gradient elution was programmed as follows: 0–3.5 min, 40% to 95% B (acetonitrile), 3.5–7.0 min 95% B, 7.1–10 min, 40% B.

The injection volume was 5 μL, the column temperature was set to 35 °C.

To detect all three available metabolites with at least two characteristic mass transitions per analyte in one run, the electrospray source of the tandem mass spectrometer operated simultaneously in positive and negative ion mode (Table 1).

Mass transitions, declustering potentials and collision energies for the compounds were obtained by direct infusion experiments with single compound standard solutions.

The following mass spectrometer settings were used: Curtain gas 40 psi, source temperature 450 °C, ion spray 4500 V, −4500 V respectively, Gas 1/Gas2: 40/60 psi.

**Preparation of analytical standards**. Stock solutions were prepared with a concentration of 100 mg/L using acetonitrile. By diluting a mix standard solution (concentration 10 mg/L) calibration standards were prepared with acetonitrile/water (60/40) at levels from 2,0 µg/L to 100 µg/L. While the stock solutions were stored at −18 °C, the calibration standards had to be newly made for each analytical run.

**Quantification of metabolites with LC-MSMS**. Due to a substantial peak separation, the identity of peaks could be confirmed using retention times and mass transitions[9].

Data were acquired and processed on Analyst 1.6.3. Quantification was carried out with an external calibration. The calibration curves were calculated by linear regression with a RSD of <20% and a coefficient of determination (R2) of >0.99.

**Statistics and reproducibility**. All biodegradation assays were performed in biological triplicates. The dot-plot format of Fig. 3 was built using the software Interactive Dot Plot Tool[20].

**Reporting summary**. Further information on research design is available in the Nature Research Reporting Summary linked to this article.

## Data availability
All data are shown in the figures and tables. Raw data generated during this study are available from the corresponding author on reasonable request. All *Mycobacteria* and *Gordonia* strains tested in this study were purchased at the German collection of Microorganisms and Cell cultures GmbH (DSMZ).

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

## Acknowledgements
The authors thank Andreas Warnstedt for kindly providing environmental samples. Further, thanks are due to Vladimir N. Belov for supply of synthesized standards of octocrylene metabolites.

## Author contributions
M.S. performed all microbiological experiments; A.Si. and A.St. carried out the mass spectrometry analyses; M.S., C.S. and H.F. drafted the paper. M.S., C.S., H.F., M.K., and G.A. designed the experimental setup.

## Competing interests
The authors declare no competing interests.
