## [Peer Review File · Communications Biology]

Reviewers' comments:

Reviewer #1 (Remarks to the Author):

The paper from Suleiman et al., describes the isolation of two bacterial strains able to degrade octocrylene (OC), a known component of sunscreens used as UV filter and responsible of pollution of both terrestrial and marine environments.

The identification of *Mycobacterium agri* and *Gordonia cholesterolivorans* strains was the result of selection experiments performed in lab from a terrestrial sample inoculated in a medium containing OC. Selection led to biofilms containing the two strains characterized by DGGE analysis. The isolation of the microorganism was unsuccessful, but when isolated *Mycobacterium* and *Gordonia* strains were used to check their ability to degrade OC, the former produced a 20% decrease in the concentration of the pollutant.

The paper is very interesting and exploit an innovative approach of enrichment on a toxic chemical. The experimental work was performed in a correct way and well explained. The results are clear and the discussion adequate.

Before publication, the author should consider the following points:

Line 89: Did the authors investigate further (by DGGE or next generation sequencing) on the nature of the mixed cultures isolated from the biofilms?

Figure 3 Did they make any effort to identify the new metabolite described?

Lines 155-158: The lack of quantitative correlation between OC and its metabolites identified by LC-MS/MS analysis was explained with a further modification of primary catabolites of OC made by *M. agri*. The authors should expand this part to make it more convincing. Which are the products of degradation of OC by non biotic factors?

Lines 163-174: The authors should comment also on marine environments which are often endangered by chemicals from sunscreens.

Minor points

Abstract: I'd change '...and present in...' with '...found in...'

Line 66: Do the authors have an idea on the origin of OC in that specific sampling site?

Fig 1A:

The structure of OC could help non-specialists

The close-up does not allow to distinguish the droplet and the biofilm. Maybe a composed picture of the incubation bottle with the close-up of the droplet and the biofilm and arrows to highlight those could be of help.

Line 221 and beyond: use italics for the microorganism names

Reviewer #2 (Remarks to the Author):

An environmental sample (water and soil) taken from a landfill site was analysed with the aim of searching for microorganisms potentially degrading the UV filter octocrylene. Two microorganisms (*Mycobacterium agri* and *Gordonia cholesterolivorans*) were tested for their ability to degrade octocrylene.

I suggest to change the title into: Microbial biofilm formation on the UV filter octocrylene and biodegradation assays.

The Authors state that results obtained by incubating with octocrylene the enrichment cultures made with two heterotrophic bacterial isolates (*Mycobacterium agri* and *Gordonia cholesterolivorans*) confirm the ability of these microorganisms to degrade this synthetic compound. The subject of this study is quite original and the results are novel. The objective is clearly indicated, the description of the materials and methods is sufficiently informative to allow replication of the experiment. Moreover, the results are consistent with the objective, the data are technically sound and the conclusions of the study are globally supported by experimental evidences, however, some suggestions or criticisms are moved to the current version of this manuscript, and I hope them would help to improve the scientific quality and technical soundness of the work, as listed below:

- 1) More details on the composition of the modified DSM media 645 and 65 should be given in the Materials section
- 2) I suggest to show graphically the results of octocrylene degradation in the presence of *M. agri* (that in the text are only shortly cited and not shown, see lines 131-133).
- 3) Data regarding the metabolites produced by *M. agri* are only cited in the text but not shown in a Figure, this would strengthen the paper further
- 4) I think that the discussion of the experimental results should be have more breath, i.e. focusing on the metabolism of the selected strains *Mycobacterium agri* and *Gordonia cholesterolivorans*, as well as on their potential implications for bioremediation. Indeed, a new eco-friendly perspective - based on the use of microbial consortia specialized in chemicals decomposition - could be potentially proposed for bioremediation of octocrylene.
- 5) Regarding the cited references, it should be better to limit references to papers strictly pertinent to the subject of this study (octocrylene) without reporting papers related to hydrothermal vents [see references 10 and 19]; this means that even for methodological procedures the Authors should select papers regarding the specific material subject of this study - octocrylene or similar synthetic compounds)
- 6) Some minor considerations:
 - a) The labels on x and y- axes in figure 3(especially for GC-MS mass spectra shown in figures D and E) are not visible, please improve the graphical resolution of these spectra;
 - b)A legend is lacking for the Table reported on pag. 13.
 - c)No data availability statement is included in this manuscript.
 - d)No statistical analysis of the data has been performed in this study, except for standard deviation/error from triplicate measurements
 - e)There are no ethical concerns arising from the use of animals or human subjects
 - f) Giving that this study seems to be original, no comparison with similar results obtained by previous studies has been included in the manuscript.
- 7) pag.2, line 34, I suggest to change "This means that" into "This relies on the possibility that suitable and efficient strains.."
pag.3, line 69, and pag. 9, line 195, subsequent streakings
pag. 4, line 85, degradation of pollutants
line 91, change "comprising" into "showing"
line 96, *Gordonia*, whose 16S rRNA
Caption to Fig. 2: observation of the biofilm formation on the OC droplet after 10 day-incubation with *G. cholesterolivorans* at 30°C and *M. agri* at 37°C
pag.6, line 138, what are "our" GC-MS spectra? those obtained from octocrylene?
pag. 8, line 161 biofilm formation is not coupled automatically with transformation or degradation
I suggest to move sentence lines 168-169 (This work highlights the potential...) to line 166, before the sentence "*M. agri* represents a potential candidate..."
line 172, by microorganisms in natural sediments and waters.
pag. 9, line 207, DNA was extracted from the gel...
pag. 10, line 211, showing within their 16S rRNA genes high similarities; line 221, *M agri* and *G. cholesterolivorans* in italics

line 226, Incubation was performed

line 229, reference standard was purchased

pag.11, line 243, The whole sample (please specify if the sample is water or soil slurry, or enrichment). if it is soil slurry, please indicate how it was obtained (i.e. dilution of volume into sterile water or physiological solution)

pag. 16, line 374, *Gordonia* in italics

Rebuttal letter

Response to Reviewer

Reviewer

The paper from Suleiman et al., describes the isolation of two bacterial strains able to degrade octocrylene (OC), a known component of sunscreens used as UV filter and responsible of pollution of both terrestrial and marine environments.

The identification of *Mycobacterium agri* and *Gordonia cholesterolivorans* strains was the result of selection experiments performed in lab from a terrestrial sample inoculated in a medium containing OC. Selection led to biofilms containing the two strains characterized by DGGE analysis. The isolation of the microorganism was unsuccessful, but when isolated *Mycobacterium* and *Gordonia* strains were used to check their ability to degrade OC, the former produced a 20% decrease in the concentration of the pollutant.

The paper is very interesting and exploit an innovative approach of enrichment on a toxic chemical. The experimental work was performed in a correct way and well explained. The results are clear and the discussion adequate.

Before publication, the author should consider the following points:

Line 89: Did the authors investigate further (by DGGE or next generation sequencing) on the nature of the mixed cultures isolated from the biofilms?

The microbial community of the enrichment culture including the biofilm was analyzed based on the DGGE method. The microbial community was composed of just 4 genera and all could be identified. Therefore, no further analysis like next generation sequencing was necessary. This is stated in line 71-76.

Figure 3 Did they make any effort to identify the new metabolite described?

Thank you, yes, we tried to analyse and identify the new metabolite but unfortunately it was not possible. We compared the MS spectra of the novel metabolite (given in figure 4) with all MS spectra available from previous work of MS spectra of putative OC metabolites found in the work of Bury et al., 2019. However, it was not possible to identify the new metabolite with the given metabolites. We added this to line 150-152. We added also a new reference for the publication of Bury et al 2019 [15].

Lines 155-158: The lack of quantitative correlation between OC and its metabolites identified by LC-MS/MS analysis was explained with a further modification of primary catabolites of OC made by *M. agri*. The authors should expand this part to make it more convincing. Which are the products of degradation of OC by non biotic factors?

Thank you for this comment. We explained this in lines 164-170. Products of non-biotic factors of OC are not known so far. However, we can exclude degradation through non-biotic factors since the control without *M. agri* did not show any degradation of OC. We added this to the lines 168-170.

Lines 163-174: The authors should comment also on marine environments which are often endangered by chemicals from sunscreens.

Thank you, we added this to line 175-177.

Minor points

Abstract: I'd change '...and present in...' with '...found in...'

Changed

Line 66: Do the authors have an idea on the origin of OC in that specific sampling site?

No, it was not possible to measure a potential OC concentration of the sampling site due the sites contained mud, plastics and other components.

Fig 1A:

The structure of OC could help non-specialists

Thank you, we added the structure of OC to figure 1.

The close-up does not allow to distinguish the droplet and the biofilm. Maybe a composed picture of the incubation bottle with the close-up of the droplet and the biofilm and arrows to highlight those could be of help.

Thank you for this advice. We added arrows to highlight the biofilm on the droplet.

Line 221 and beyond: use italics for the microorganism names

Thank you; changed accordingly.

Reviewer #2 (Remarks to the Author):

An environmental sample (water and soil) taken from a landfill site was analysed with the aim of searching for microorganisms potentially degrading the UV filter octocrylene. Two microorganisms (*Mycobacterium agri* and *Gordonia cholesterolivorans*) were tested for their ability to degrade octocrylene.

I suggest to change the title into: Microbial biofilm formation on the UV filter octocrylene and biodegradation assays.

We agree. Thank you for the constructive comment.

The Authors state that results obtained by incubating with octocrylene the enrichment cultures made with two heterotrophic bacterial isolates (*Mycobacterium agri* and *Gordonia cholesterolivorans*) confirm the ability of these microorganisms to degrade this synthetic compound. The subject of this study is quite original and the results are novel. The objective is clearly indicated, the description of the materials and methods is sufficiently informative to allow replication of the experiment. Moreover, the results are consistent with the objective, the data are technically sound and the conclusions of the study are globally supported by experimental evidences, however, some suggestions or criticisms are moved to the current version of this manuscript, and I

hope them would help to improve the scientific quality and technical soundness of the work, as listed below:

1) More details on the composition of the modified DSM media 645 and 65 should be given in the Materials section

Thank you, details are added in line 225-233.

2) I suggest to show graphically the results of octocrylene degradation in the presence of *M. agri* (that in the text are only shortly cited and not shown, see lines 131-133).

Thank you for this advice. A figure (Fig. 3) was added to the manuscript.

3) Data regarding the metabolites produced by *M. agri* are only cited in the text but not shown in a Figure, this would strengthen the paper further

Since the metabolites did not occur in a high concentration ($\mu\text{g/L}$ instead of mg/L for octocrylene degradation) we just mentioned the data in the text without showing them in a figure. As we discussed in line 164-170, the occurrence of the metabolite concentration does not go along with the decrease of octocrylene. Therefore, *M. agri* probably uses metabolites of octocrylene for growth.

4) I think that the discussion of the experimental results should be have more breath, i.e. focusing on the metabolism of the selected strains *Mycobacterium agri* and *Gordonia cholesterolivorans*, as well as on their potential implications for bioremediation. Indeed, a new eco-friendly perspective - based on the use of microbial consortia specialized in chemicals decomposition - could be potentially proposed for bioremediation of octocrylene.

Thank you for this advice. We added more text to the discussion part in line 179-183.

5) Regarding the cited references, it should be better to limit references to papers strictly pertinent to the subject of this study (octocrylene) without reporting papers related to hydrothermal vents [see references 10 and 19]; this means that even for methodological procedures the Authors should select papers regarding the specific material subject of this study - octocrylene or similar synthetic compounds)

Thank you for this advice, we removed the reference number 19 (the old one).

Unfortunately, we cannot remove the reference 10, because the microbial diversity analysis was performed according to the methodological procedures reported in this paper. Furthermore, the amount of papers reporting the degradation of OC with diversity analysis is pretty low and none of these performed our analysis.

6) Some minor considerations:

a) The labels on x and y- axes in figure 3(especially for GC-MS mass spectra shown in figures D and E) are not visible, please improve the graphical resolution of these spectra;

Thank you, the quality has improved very much

b)A legend is lacking for the Table reported on pag. 13.

Thank you, legend was added

c) No data availability statement is included in this manuscript.

Was added

d) No statistical analysis of the data has been performed in this study, except for standard deviation/error from triplicate measurements

No other statistical analyses were necessary in this proof-of-concept study.

e) There are no ethical concerns arising from the use of animals or human subjects

Was added

f) Giving that this study seems to be original, no comparison with similar results obtained by previous studies has been included in the manuscript.

This is, to our knowledge, the first study dealing with the bacterial degradation of OC.

7) pag.2, line 34, I suggest to change "This means that" into "This relies on the possibility that suitable and efficient strains.."

Changed

pag.3, line 69, and pag. 9, line 195, subsequent streakings

Changed

pag. 4, line 85, degradation of pollutants

Changed

line 91, change "comprising" into "showing"

Changed

line 96, *Gordonia*, whose 16S rRNA

Changed

Caption to Fig. 2: observation of the biofilm formation on the OC droplet after 10 day-incubation with *G. cholesterolivorans* at 30°C and *M. agri* at 37°C

Changed

pag.6, line 138, what are "our" GC-MS spectra? those obtained from octocrylene?

To be more clear we have changed the sentence: "GC-MS spectra obtained in this work with the metabolite`s GC-MS spectra and LC-MS/MS which were subsequently used to..."

pag. 8, line 161 biofilm formation is not coupled automatically with transformation or degradation

I suggest to move sentence lines 168-169 (This work highlights the potential...) to line 166, before the sentence "M. agri represents a potential candidate..."

Thank you for this suggestion, but we think it fits good at the former place, since it is dealing with *G. cholesterolivorans* „Interestingly, the analysis of growth experiments with *G. cholesterolivorans* revealed no change of OC concentration during incubation, although biofilm was observed". → biofilm formation is not coupled automatically...

line 172, by microorganisms in natural sediments and waters.

Changed

pag. 9, line 207, DNA was extracted from the gel...

Changed

pag. 10, line 211, showing within their 16S rRNA genes high similarities; line 221, *M agri* and *G. cholesterolivorans* in italics

Changed

line 226, Incubation was performed

Changed

line 229, reference standard was purchased

Changed

pag.11, line 243, The whole sample (please specify if the sample is water or soil slurry, or enrichment). if it is soil slurry, please indicate how it was obtained (i.e. dilution of volume into sterile water or physiological solution)

Sorry for the confusion. The „whole sample“ was replaced with „incubation culture of *M. agri*/*G. cholesterolivorans*“ which was used for the assay (and the controls without bacteria). This was added to line 259.

pag. 16, line 374, *Gordonia* in italics

Changed